# Elevated serum magnesium lowers calcification propensity in *Memo1*-deficient mice

Matthias B. Moor[1,2☯¤], Suresh K. Ramakrishnan[1,2☯], Finola Legrand[1],
Matthias Bachtler[3], Robert Koesters[4], Nancy E. Hynes[5], Andreas Pasch[3],
Olivier Bonny[1,2,6]*

**1** Department of Medical Biosciences, University of Lausanne, Lausanne, Switzerland, **2** The National Centre of Competence in Research (NCCR) "Kidney.CH - Kidney Control of Homeostasis", Zürich, Switzerland, **3** Calciscon AG, Nidau, Switzerland and Institute for Physiology and Pathophysiology, Johannes Kepler University Linz, Linz, Austria, **4** Department of Nephrology, Hôpital Tenon, Université Pierre et Marie Curie, Paris, France, **5** Friedrich Miescher Institute for Biomedical Research and University of Basel, Basel, Switzerland, **6** Department of Medicine, Service of Nephrology, Lausanne University Hospital, Lausanne, Switzerland

☯ These authors contributed equally to this work.
¤ Current address: Department of Nephrology and Hypertension, Bern University Hospital, Bern, Switzerland
* Olivier.Bonny@unil.ch

**Data Availability Statement:** All relevant data are within the paper and its Supporting Information files.

## Abstract

MEdiator of cell MOtility1 (MEMO1) is a ubiquitously expressed redox protein involved in extracellular ligand-induced cell signaling. We previously reported that inducible whole-body *Memo1* KO (cKO) mice displayed a syndrome of premature aging and disturbed mineral metabolism partially recapitulating the phenotype observed in *Klotho* or *Fgf23*-deficient mouse models. Here, we aimed at delineating the contribution of systemic mineral load on the *Memo1* cKO mouse phenotype. We attempted to rescue the *Memo1* cKO phenotype by depleting phosphate or vitamin D from the diet, but did not observe any effect on survival. However, we noticed that, by contrast to *Klotho* or *Fgf23*-deficient mouse models, *Memo1* cKO mice did not present any soft-tissue calcifications and displayed even a decreased serum calcification propensity. We identified higher serum magnesium levels as the main cause of protection against calcifications. Expression of genes encoding intestinal and renal magnesium channels and the regulator epidermal growth factor were increased in *Memo1* cKO. In order to check whether magnesium reabsorption in the kidney alone was driving the higher magnesemia, we generated a kidney-specific *Memo1* KO (kKO) mouse model. *Memo1* kKO mice also displayed higher magnesemia and increased renal magnesium channel gene expression. Collectively, these data identify MEMO1 as a novel regulator of magnesium homeostasis and systemic calcification propensity, by regulating expression of the main magnesium channels.

**Funding:** The study was sponsored by the Swiss National Science Foundation through the program NCCR-Kidney.CH (183774) and by an unrestricted grant from the patient association "Association pour l'Information et la Recherche sur les maladies rénales Génétiques (AIRG)-Suisse". The funders had no role in study design, data collection and analysis, decision to publish, or preparation of the manuscript.

**Competing interests:** AP is an inventor of the T50-Test and stock-holder and employee of Calciscon AG, which commercialized the T50-Test (calcification propensity test). MB is an employee of Calciscon AG. This does not affect our adherence to PLOS ONE policies on data or materials sharing.

# Introduction

MEMO1 is a redox protein [1] that acts in different signaling pathways including FGF23—FGF receptor [2]. Embryonic lethality results from constitutive deletion of *Memo1* [3], but mice conditionally lacking *Memo1* exon 2 (*Memo1* cKO) showed, after post-natal Cre-lox-mediated excision, a phenotype of premature aging and disturbed mineral homeostasis [2]. More precisely *Memo1* cKO mice on FVBxC57BL/6 mixed genetic background displayed growth retardation, alopecia, insulin hypersensitivity, and subcutaneous fat atrophy. More-over, mineral homeostasis was affected, with elevated calcemia and 1,25-$(OH)_2$-vitamin $D_3$ serum levels, slightly higher FGF23 serum concentrations and reduced parathyroid hormone (PTH) levels compared to controls [2]. *Memo1* cKO mice backcrossed to C57BL/6 background showed shortened life span, a spatially heterogeneous bone phenotype consisting of osteopenia and hyperostosis, elevated calcemia and FGF23 serum concentrations, but normal serum concentrations of phosphate, 1,25-$(OH)_2$-vitamin $D_3$ and PTH [4].

Strikingly, the triad of premature aging, insulin hypersensitivity and altered mineral homeostasis identified in *Memo1* cKO mice substantially overlaps with the phenotypes of *Klotho*-deficient [5–7] or *Fgf23* KO mice in which altered phosphate and vitamin D metabolism are the primary triggers of aging [8]. Together with results obtained from several in vitro and in vivo experiments [2], these data suggest that MEMO1 may participate in the FGF23-KLOTHO-FGFR signaling axis.

Humans carrying loss-of-function variants of *FGF23* or *KLOTHO* suffer from familial tumoral calcinosis, a condition in which affected subjects develop ectopic calcifications. Like-wise, both *Klotho*-deficient [5] and *Fgf23* KO mice [9] showed extensive soft tissue calcifications. Genetic or dietary interventions that suppressed the vitamin D axis [9–15] or considerably decreased the phosphate load [9, 16, 17] prevented premature aging, soft-tissue calcifications and other traits to a variable extent in these two mouse models.

Based on the similarities between the phenotype of the *Memo1* cKO mice and that of *Fgf23*- or *Klotho*-deficient strains, we tested the hypothesis that the *Memo1* cKO phenotype might be mediated by disturbed mineral metabolism and ectopic calcifications. We looked for soft tissue calcifications and subjected the mice to low phosphate diet or to vitamin-D deficient diet. We found that by contrast to *Fgf23*- or *Klotho*-deficient mouse models, the *Memo1* cKO mice phenotype is not influenced by the low phosphate or vitamin-D deficient diets. Further, these mice exhibit no soft tissue calcifications, and display lower serum calcification propensity. We found that this lower calcification propensity was due to higher magnesemia and showed that gene expression of *Trpm6* and *Trpm7*, two trans-epithelial magnesium channels, are upregulated in absence of *Memo1* and involved in the altered magnesium homeostasis.

# Methods

## Animal studies

All animal experiments were conducted as approved by the veterinary service of the Canton de Vaud, Switzerland.

Inducible whole-body conditional *Memo1* KO (cKO) and inducible kidney-specific *Memo1* KO (kKO) mouse models on C57BL/6 genetic background were generated as previously described [4] using an allele with *Memo1* exon 2 flanked by loxP sites [2]. In brief, *pCX-CreER*$^{TM}$/*Memo1*$^{fl/fl}$ mice were treated by 3 intraperitoneal injections with 2mg tamoxifen (Sigma T5648) at 4 or 8 weeks of age, as indicated, to obtain postnatal *Memo1* deletion in *Memo1* cKO mice. *Memo1*$^{fl/fl}$ littermates not carrying Cre underwent identical treatment to serve as controls.

For generation of *Memo1* kKO mice, B6.Cg-Tg(Pax8 rtTA2S\*M2)1Koes/J [18] and LC-1 transgene under a Ptet bi-1 promoter [19] were combined with the *Memo1*^fl/fl^ allele. The resulting males were treated for 14 days with low-dose 0.2mg/mL doxycycline hyclate (Sigma) in 2% sucrose in tap water starting at age 25 to 30 days. Male *Memo1*^fl/fl^ littermates lacking at least 1 of the other transgenes were treated with the same doxycycline dose and served as controls.

Both models were genotyped using ear punch biopsy or tail DNA using the primers: *Memo1* forward 5'-*CCCTCTCATCTGGCTTGGTA*-3', *Memo1* reverse 5'- *GCTGCATATG CTCACAAAGG*-3', *Cre* forward 5'-*AGGTTCGTGCACTCATGGA*-3', *Cre* reverse 5'-*TC ACCAGTTTAGTTACCC*-3', Pax8 rtTA forward 5'-*CCATGTCTAGACTGGACAAGA*-3', Pax8 rtTA reverse 5'-*CTCCAGGCCACATATGATTAG*-3'.

Mice were maintained on standard laboratory chow containing 1% calcium, 0.65% phosphorus, 0.23% magnesium and vitamin D 1600 IU/kg, unless stated otherwise. For experimental induction of vitamin D deficiency or dietary phosphate depletion, the following protocol was used:

Two to five mice from an entire cage were randomized to receive either a control diet or a dedicated diet. Body weight of the mice was initially monitored 1–2 times weekly for phosphate depletion experiments and twice weekly for vitamin D depletion experiments. If a specific phenotype was observed, mice were followed more closely, every 1–2 days. Health status was checked at least twice weekly. End points for each mouse used in these experiments were met if at least one of the following criteria was fulfilled: 1) Loss of 10% of body weight in two consecutive measurements, 2) Body weight falling below 85% of average control littermates on the same diet, 3) Failure to thrive, move or eat, 4) Signs of distress, 5) Death. Observation of criteria 1 to 4 was followed by euthanasia.

The phosphate depletion diet contained 0.2% phosphate and was compared to a 0.8% phosphate control diet. Both diets contained, 1.2% calcium, vitamin A 4000 IU/kg, vitamin D 1000 IU/kg, vitamin E 100 mg/kg, protein 18%, crude fat 7%, lysine 14g/kg (Kliba, #2222, Switzerland) and were introduced after weaning, 2–3 days prior to tamoxifen treatment.

Vitamin D depletion experiments were performed with a diet containing vitamin D 0 IU/kg (Altromin GmbH C1017; calcium 0.95%, phosphorus 0.8%, magnesium 0.07%, vitamin A 15000 IU/kg, vitamin E 164mg/kg, protein 18%, crude fat 5%, crude fiber 4%, lysine 1.7%) or a control diet containing 500 IU/kg vitamin D (Altromin GmbH C1000; calcium 0.93%, phosphorus 0.8%, magnesium 0.07%, vitamin D 500 IU/kg, vitamin A 14000 IU/kg, vitamin E 164mg/kg, protein 18%, crude fat 5%, crude fiber 4%, lysine 1.7%). To minimize exposure to vitamin D, diets were introduced to parental cages prior to gestation of experimental mice, after randomization.

For collection of 24 hour urine samples, metabolic cages 3600M021 (Tecniplast) were used to which mice were accustomed for 2 days prior to measurements. Mice had free access to food and water. Mice were deeply anesthetized with ketamine/xylazine and bled by orbital puncture followed by cervical dislocation.

## Chemical analyses

For serum and urinary analyses, specimens from several animals were combined to obtain biologically independent pools, as indicated; for *Memo1* cKO and controls, specimens from 4 individual males were pooled; for *Memo1* kKO and controls, specimens from 2 males and 2 females were combined.

Serum and urinary electrolytes were quantified by the Lausanne University Hospital central laboratory: Magnesium was measured by the xylidyl blue method, total calcium by the

NM-BAPTA method, phosphate by the phosphomolybdate method, and creatinine by the modified Jaffé method. Tissue calcium content of murine thoracic aorta was measured after drying the samples at 60˚C for 20h and eluted in HCl 1M over 48h using a chromogenic o-cresophthalein kit (Sigma MAK022). T50 test of calciprotein particle conversion was measured by Calciscon AG, Nidau, Switzerland, as described [20]. For magnesium spiking experiments, magnesium chloride was added to samples as indicated prior to performing T50 measurements.

## Gene expression

RNA was extracted using TRI reagent (Applied Biosystems by Life Technologies) and concentration was measured by Nanodrop (Nanodrop 2000, Thermo Fisher Scientific, Waltham, MA, USA). RNA was reverse transcribed using PrimeScript RT (Takara Bio Inc, Otsu, Japan). We performed qPCR by using the SYBR Green method (Applied Biosystems by Life Technologies) on a 7500 Fast machine (Applied Biosystems). Samples were run as triplicates, and *Actb* or *Gapdh* were used as house-keeping genes for relative quantification performed by using the delta-delta CT method. For each run, melting curves were obtained in order to verify the specificity of the signal and amplified products were visualized on agarose gel. Primers were purchased from Microsynth (Switzerland), and sequences are shown in Table 1.

## Protein isolation

Tissue proteins were extracted in NP-40 buffer (50 mM HEPES pH7.4; 150mM NaCl; 25mM NaF; 1mM EDTA; 5mM EGTA; 1% Nonidet P-40; 2M Na ortho-vandate, and 1mM DTT supplied with 10 µg/L leupeptin (Applichem by Axonlab), 10 µg/L aprotinin, 1mM PMSF) and lyzed by metal beads. Homogenates were spun down. For preparation of membrane protein enriched fractions, kidneys were homogenated by a Polytron (Kinematica AG, Switzerland) in sucrose buffer (250mM sucrose, 150mM NaCl, 30mM Tris pH7.5, 1mM PMSF, 10 µg/L aprotinin, 10 µg/L leupeptin, and Pepstatin) and spun down 2x10min at 1000g. Supernatants were spun down again at 100'000g for 1h; pellets were resuspended in 75uL sucrose buffer. Twenty to 50ug of protein was denaturated in Laemmli buffer containing beta-mercaptoethanol. Proteins were separated by 7%, 10% or 13% SDS-PAGE, transferred to nitrocellulose (PROTRAN, Whatman) or PVDF (BioRad) membranes and visualized using Ponceau S. After blocking in nonfat dried milk 5%-TBST, membranes were incubated with primary antibodies detecting MEMO1 1:2000 [2], NCX1 1:1000 [21], TRPV5 1:50 [22], anti-pan PMCA 1:500 (Sigma A7952), calbindin D28K 1:1000 (Sigma C7354), NAPI2A 1:4000 [23] (gift from Carsten Wagner), ACTIN 1:2000 (Sigma A2066) followed by anti-mouse or, respectively, anti-rabbit horseradish peroxidase-conjugated secondary antibodies 1:10'000 (Milian Analytica 115-035-003 and 111-035-003) and exposed using Fusion Solo (Witec). Densitometric quantification of protein signal was obtained by ImageJ (1.48v) using ACTIN as a loading control for whole-tissue lysates and Ponceau S for membrane-enriched protein fractions.

## Histology

Sectioning and staining of paraffin-embedded kidneys was performed by the Mouse Pathology Facility of the University of Lausanne. Aorta cryosections were stained in-house using standard Von Kossa staining protocols. Bone tissue and calcified rat aorta served as positive controls.

**Table 1. Primers used for qPCR.**

| Oligonucleotide | 5'-sequence-3' |
| --- | --- |
| Slc8a1 forward | AGAGCTCGAATTCCAGAACGATG |
| Slc8a1 reverse | TTGGTTCCTCAAGCACAAGGGAG |
| Trpv5 forward | TCCTTTGTCCAGGACTACATCCCT |
| Trpv5 reverse | TCAAATGTCCCAGGGTGTTTCG |
| Calb1 forward | AACTGACAGAGATGGCCAGGTTA |
| Calb1 reverse | TGAACTCTTTCCCACACATTTTGAT |
| Atp2b4 forward | CTTAATGGACCTGCGAAAGC |
| Atp2b4 reverse | ATCTGCAGGGTTCCCAGATA |
| beta-actin forward | GTC CAC CTT CCA GCA GAT GT |
| beta-actin reverse | AGT CCG CCT AGA AGC ACT TGC |
| Gapdh forward | CCA CCC AGA AGA CTG TGG AT |
| Gapdh reverse | CAC ATT GGG GGT AGG AAC AC |
| Slc34a1 forward | TCACAGTCTCATTCGGATTTGG |
| Slc34a1 reverse | GGCCTCTACCCTGGACATAGAA |
| Slc34a3 forward | CCTACCCCCTCTTCTTGGGT |
| Slc34a3 reverse | AGAGCAACCTGAACTGCGAA |
| Cyp27b1 forward | ATGTTTGCCTTTGCCCAGA |
| Cyp27b1 reverse | GACGGCATATCCTCCTCAGG |
| Cyp24a1 forward | GAAGATGTGAGGAATATGCCCTATTT |
| Cyp24a1 reverse | CCGAGTTGTGAATGGCACACT |
| Ahsg forward | CGACAAAGTCAAGGTGTGGTC |
| Ahsg reverse | TCAGCTGCCTCACAGAACAGT |
| Egf forward | GAGTTGCCCTGACTCTACCG |
| Egf reverse | CCACCATTGAGGCAGTATCC |
| Egfr forward | CAGAACTGGGCTTAGGGAAC |
| Egfr reverse | GGACGATGTCCCTCCACTG |
| Trpm6 forward | AAAGCCATGCGAGTTATCAGC |
| Trpm6 reverse | CTTCACAATGAAAACCTGCCC |
| Trpm7 forward | GGTTCCTCCTGTGGTGCCTT |
| Trpm7 reverse | CCCCATGTCGTCTCTGTCGT |

## Tissue calcification imaging

For *ex vivo* analysis of whole-animal tissue calcification, a soft X-ray scout-scan mode of a Sky-Scan 1076 micro-computed tomography machine (Skyscan, Kontich, Belgium) was employed.

## Copper reduction assay

Recombinant full-length MEMO1 protein produced in *E. coli* was purchased from antibodies-online.com (ABIN2130536). Ascorbic acid was purchased from Fluka. An OxiSelect™ total antioxidant capacity assay (Cell Biolabs, STA-360) was used for copper-(II) reduction assay according to manufacturer's instructions using uric acid standards. $MgCl_2$ was added to recombinant MEMO1 protein as indicated, and ascorbic acid was used as a positive control. All samples were run in 20μL total sample volume and in duplicates.

## Data analysis

Data from experiments with 2 independent groups were analyzed by unpaired t-test. For analysis of 2 sources of variability in 4 experimental groups and their interaction (effects of

genotype, diet, and interaction between genotype and diet), two-way ANOVA were calculated using GraphPad PRISM 5.03. A power analysis was performed using web-based software at https://www.statstodo.com/SSizSurvival_Pgm.php (accessed on February 10th, 2015). To detect a difference in survival rates from 0.01 to 0.5, a total of 10 animals per group was required for a type I error of 0.05 and a power of 0.8. Data from survival studies were analyzed by Log Rank (Mantel Cox) test which assumes a constant hazard ratio at all time points. Two-sided p-values <0.05 were considered significant.

## Results

### Altered renal calcium handling in *Memo1* cKO mice

We backcrossed and validated the previously described [2] *Memo1* cKO mouse model on a pure C57BL/6 background. We have previously described an elevated calcemia and FGF23 concentration in the serum of these mice, in addition to increased urinary calcium excretion over 24h [4]. Now, we characterized transport proteins involved in mineral homeostasis at a molecular level to delineate similarities or differences between the *Memo1*-deficient mice in comparison to what has been published on the phenotypically close *Klotho* or *Fgf23*—deficient models.

An assessment of the expression of the main calcium transporting molecules in the gut revealed no change between *Memo1* cKO mice and controls in the duodenum (Fig 1A). In the kidney, we measured transcription of genes encoding proteins involved in active calcium reabsorption of the tightly regulated distal convoluted tubule—connecting tubule (DCT-CNT). Expression of the *Trpv5* gene coding for the apical Transient Receptor Potential cation channel, subfamily V, member 5 (TRPV5) and *Calb1* coding for intracellular calbindin D28K were increased in *Memo1* cKO (Fig 1B). Similarly, renal transcripts of *Atp2b4* and *Slc8a1* coding for the basolaterally expressed plasma membrane $Ca^{2+}$ ATPase 4 (PMCA4) and the sodium calcium exchanger (NCX1) were also increased upon *Memo1* deletion (Fig 1B).

Next, we investigated these calcium transport molecules at the protein level. Membrane abundance of TRPV5 protein was increased three-fold in *Memo1* cKO (Fig 1C). In whole kidney lysates, the abundance of intracellular calbindin D28K protein was increased (Fig 1D). Finally, both NCX1 membrane protein abundance (Fig 1E) and PMCA membrane abundance were increased (Fig 1C). Densitometric quantifications of these Western blots are shown in Fig 1F.

### Evidence for disturbed renal phosphate transport and vitamin D metabolism in *Memo1* cKO mice

In order to further characterize the extent of disturbed mineral homeostasis, we assessed genetic markers of phosphate homeostasis in the kidney. We found that renal transcription of *Slc34a1* encoding sodium-dependent phosphate transporter (NAPI2A) tended to decrease in *Memo1* cKO, whereas transcription of *Slc34a3* encoding NAPI2C significantly diminished upon *Memo1* deletion (Fig 2A). Gene expression of *Cyp24a1* encoding the vitamin D—inactivating 24 alpha-hydroxylase was increased in *Memo1* cKO, whereas *Cyp27b1* was unchanged (Fig 2A). On a protein level, we found that renal NAPI2A membrane abundance was increased in *Memo1* cKO (Fig 2B, quantification in Fig 2C). Altogether, *Memo1* seems to modulate both transcriptional and protein expression level of several key players in mineral metabolism [2, 4].

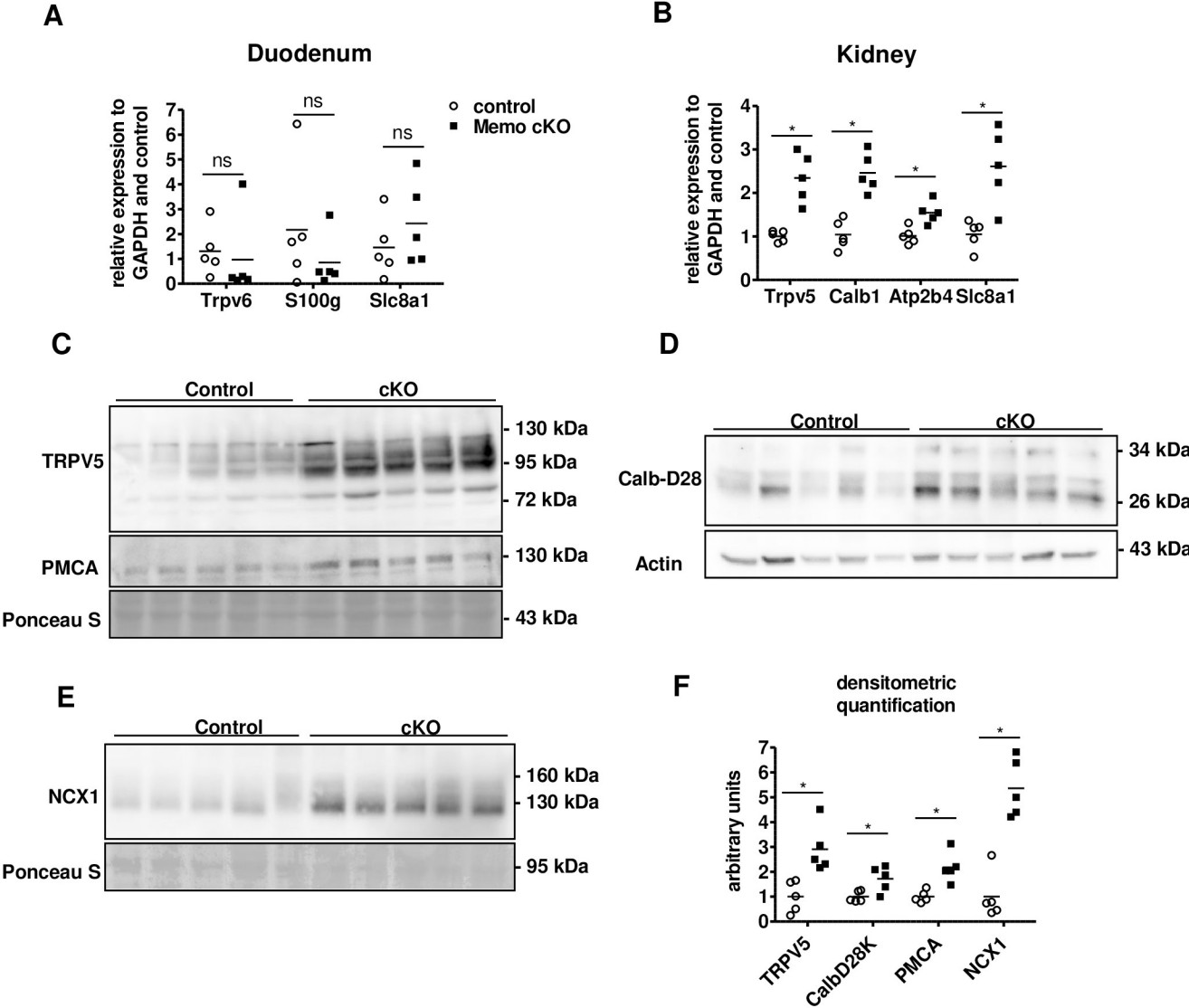

**Fig 1. Increased renal calcium transport proteins in whole-body *Memo1* cKO mice. A**. Gene expression of the duodenal calcium transport proteins *Trpv6*, *S100g* and *Slc8a1* revealed no difference between genotypes. **B**. Renal gene expression of *Trpv5*, *Calb1*, *Atp2b4 and Slc8a1* were increased in *Memo1* cKO compared to controls. **C-E**. Renal membrane protein-enriched fractions (C, E) and whole kidney lysates (D) were used for Western blotting of proteins corresponding to genes assessed in B. Calb-D28, calbindin D28K. **F**. Densitometric quantification of C-E. * p < 0.05 (t-test); n = 5 per genotype.

## No change in survival in whole-body *Memo1* cKO upon diminished systemic mineral load

We assessed the impact of the mineral load on disease-free survival in *Memo1* cKO mice by testing two different diets: a low phosphate diet and a vitamin D deficient diet. *Memo1* cKO and control mice were randomly assigned to low phosphate (0.02%) versus control diet containing 0.8% phosphate, or to 0 IE/kg versus 500 IE/kg of vitamin D containing diet. To ensure depletion of vitamin D in adipose tissue of mice under experimental vitamin D deficiency, the parents of experimental mice were fed a vitamin D deficient diet prior to gestation.

The 0.02% low phosphate diet decreased urinary phosphate excretion in both *Memo1* cKO and control mice (Fig 3A). The vitamin D- deficient diet severely reduced total serum calcium

**A**

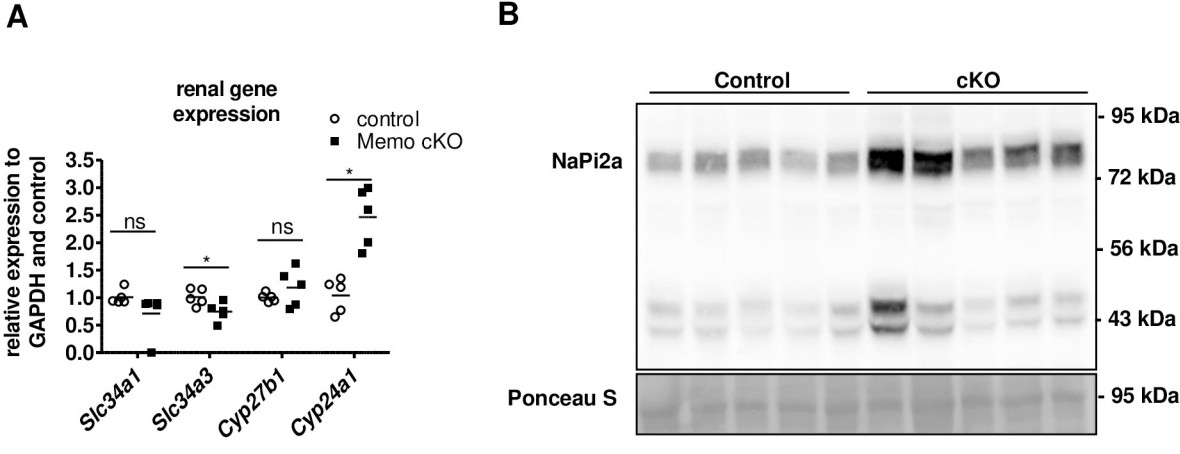

**B**

**C**

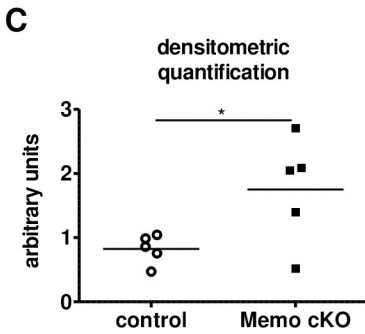

**Fig 2. Alterations in phosphate transporters expression and vitamin D metabolism in whole-body *Memo1* cKO. A**. Transcripts *Slc34a1* and *Slc34a3* coding for phosphate transporters, and genes *Cyp27b1* and *Cyp24a1* coding for vitamin D-metabolizing enzymes were assessed by kidney qPCR in controls and *Memo1* cKO. **B**. Western blot analysis showed increased membrane abundance of NAPI2a protein in *Memo1* cKO; **C**. Densitometric quantification of B. ns, not significant; * $p < 0.05$ (t-test). n = 5 per genotype.

levels of both genotypes (Fig 3B). However, none of the two interventional diets had an impact on disease-free survival of *Memo1* cKO mice compared to *Memo1* cKO mice on the corresponding control diet (Fig 3C and 3D).

Collectively, these results argue against a major influence of phosphate or vitamin D on the premature aging and death phenotype reported in *Memo1* cKO mice.

## Absence of soft-tissue mineralization in *Memo1* cKO mice

We further characterized the existence and extent of soft-tissue calcifications in *Memo1* cKO mice. We dissected the entire thoracic aorta and found that the calcium content was comparable between *Memo1* cKO mice and controls (Fig 4A). In addition, we investigated the abdominal aorta by histological analysis and von Kossa staining in control (Fig 4B) and *Memo1* cKO mice (Fig 4C). We did not notice any calcification in either genotype. Similarly, kidney sections stained with von Kossa revealed no calcification in control (Fig 4D) or *Memo1* cKO mice (Fig 4E). Finally, whole-body *ex vivo* X-ray scans revealed no soft tissue calcification in the whole body of *Memo1* cKO mice and controls (Fig 4F). Altogether, we found no evidence of tissue calcification in *Memo1* cKO mice that could explain the premature aging phenotype in contrast to what has been described for *Fgf23-* or *Klotho*-deficient mice.

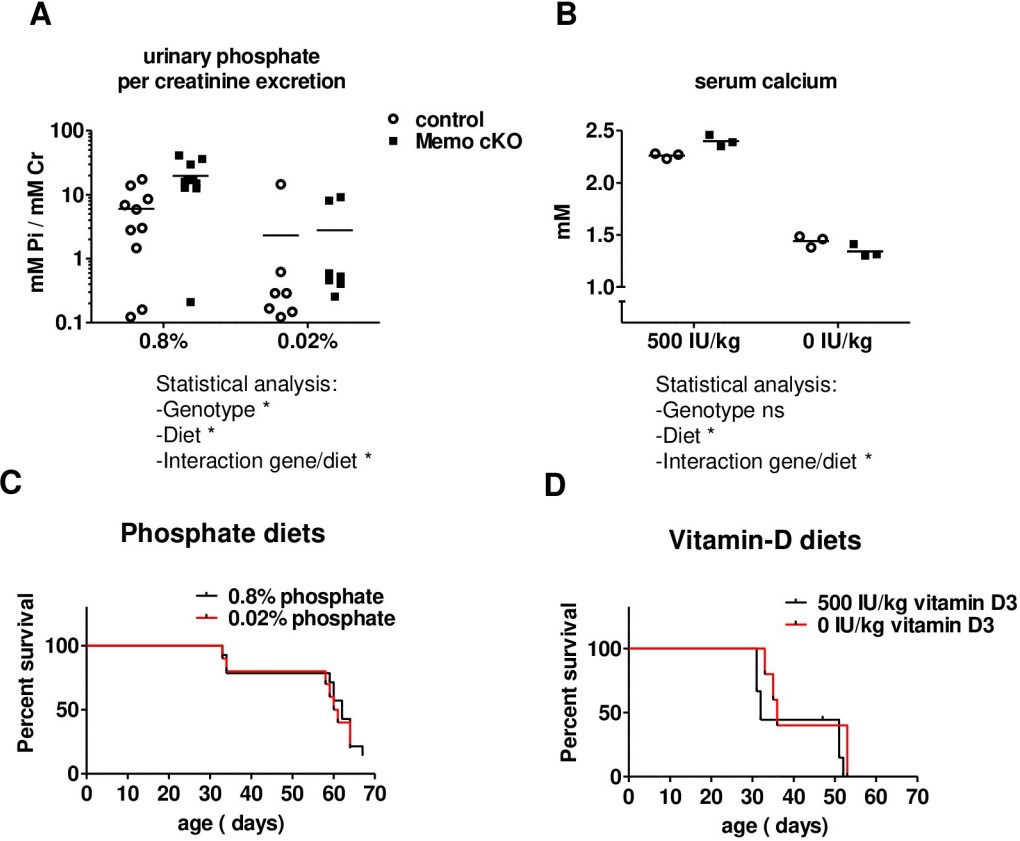

**Fig 3. Systemic mineral depletion does not affect disease-free survival of whole-body *Memo1* cKO mice. A**. Dietary phosphate depletion from 0.8 to 0.02% phosphorus content decreased urinary phosphate per creatinine excretion in both *Memo1* cKO and control mice. **B**. Mice of either genotype fed vitamin D-free diet displayed lower serum calcium than those fed a diet containing 500 IU/kg vitamin-D. **C-D**. Survival analysis revealed no differences in disease-free survival of *Memo1* cKO mice fed diets varying in phosphate (C) or vitamin D content (D). IU, international units. Statistical analysis was obtained by Two-way ANOVA (A, B) and by Log Rank test (Mantel-Cox) (C, D); ns, not significant; * p < 0.05. n = 10 per genotype on 0.8% and 7 per genotype on 0.02% phosphate diets (A). n = 3 per condition (B). 100% equals initial n at start with the following number of animals: 14 (under 0.8% Pi diet) and 10 (under 0.02% Pi diet) animals for phosphate diets (C); 9 (for the 500 IU/kg vitamin D diet) and 8 (for the 0 IU/kg vitamin D diet) animals for the vitamin D3 diets (D).

## Decreased serum calcification propensity and higher plasma magnesium concentration in *Memo1* cKO mice

The absence of soft tissue calcifications in *Memo1* cKO mice led us to investigate the overall calcification propensity in the mouse serum by nephelometry that measures conversion from spontaneously formed primary to secondary calciprotein particles (CPPs) [20]. Maximal light scattering in relative nephelometry units (RNU) was decreased in the sera of all *Memo1* cKO mice compared to controls (Fig 5A). In addition, T50 time of conversion from primary to secondary CPPs was prolonged in *Memo1* cKO mice, indicating a diminished tendency to form secondary CPPs (Fig 5A). Consequently, we assessed the presence of putative calcification inhibitors in *Memo1* cKO mice sera. We measured transcription of *Ahsg* encoding Fetuin-A by qPCR in the liver, but this was unchanged in *Memo1* cKO compared to controls (Fig 5B). Magnesium is another strong inhibitor of calcification. We pooled serum samples of 24 individual mice per genotype into 6 independent biological samples per genotype and found

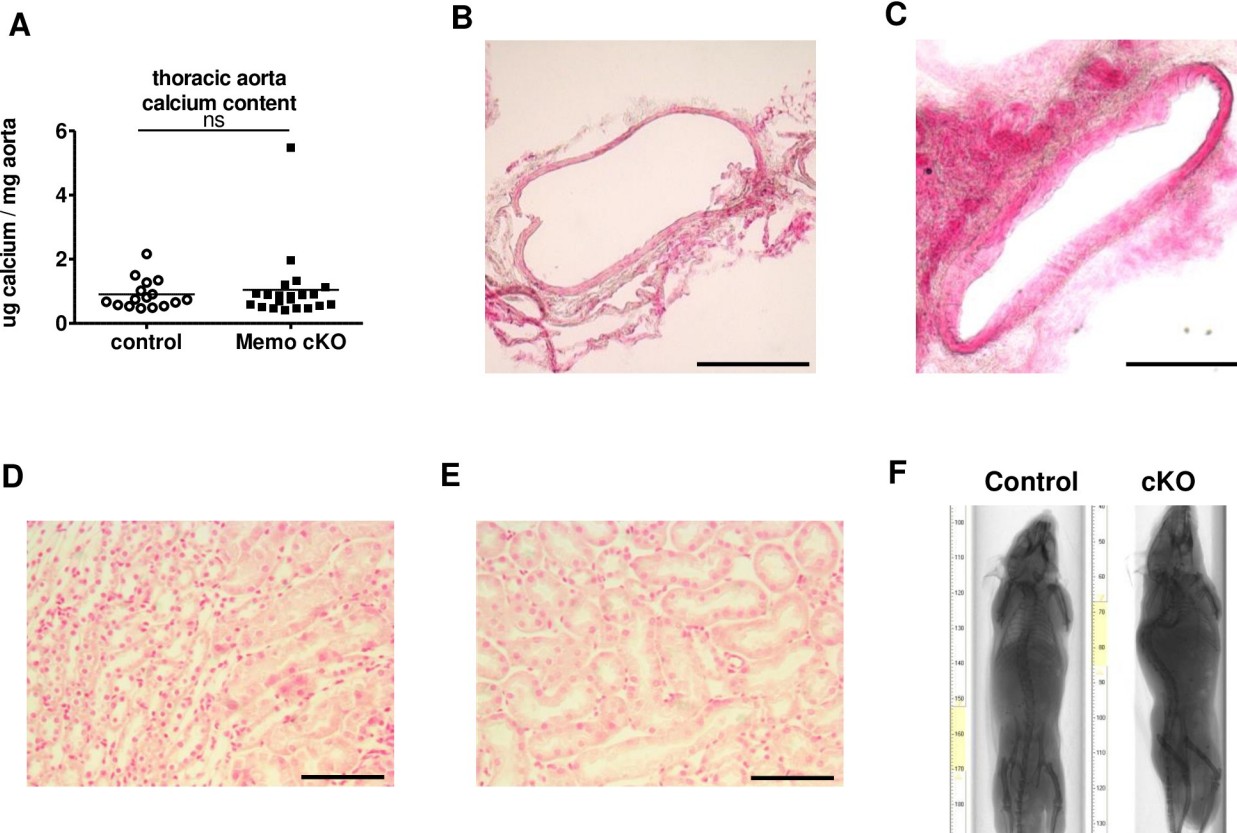

**Fig 4. Whole-body *Memo1* cKO mice do not display ectopic calcification. A**. Calcium content eluted from thoracic aortae showed no differences between *Memo1* cKO and control animals. **B-C**. Histological analysis of Von Kossa stained abdominal aortae revealed no difference between control (B) and *Memo1* cKO (C) mice. **D-E**. Renal parenchyma stained by Von Kossa showed no calcified areas in control (D) or *Memo1* cKO (E). **F**. Whole-body X-ray scans of control and *Memo1* cKO mice showed no extensive tissue calcification in either genotype. ns, not significant (t-test). Scale bars: 200 μm (B-C), 100 μm (D-E). n = 16 controls and 21 *Memo1* cKO mice (A); representative images of 2 per genotype (B-C and F) and 3 per genotype (D-E).

substantially increased serum magnesium concentrations in *Memo1* cKO compared to controls (Fig 5C) (1.07 ± 0.04 mmol/L vs 0.87 ± 0.04 mmol/L, p<0.001). Of note, urinary excretion rates of magnesium over 24h were comparable between the genotypes (Fig 5D).

To delineate the mechanisms of the observed difference in serum magnesium levels, we quantified transcripts levels of genes encoding magnesium channels in the intestine and in the kidney. Transcripts of *Trpm6* encoding Transient Receptor Potential cation channel, subfamily M, member 6 (TRPM6) but not *Trpm7* were increased in the ileum of *Memo1* cKO mice compared to controls (Fig 5E), suggesting a state of increased intestinal magnesium absorption through TRPM6 in the gut. In the kidney of *Memo1* cKO mice, both *Trpm6* and *Trpm7* expression were increased (Fig 5F). As renal magnesium handling is influenced by epidermal growth factor receptor (EGFR) signaling [24], we investigated transcriptional levels of the genes *Egf* and *Egfr*. Renal *Egf* was increased in *Memo1* cKO, but *Egfr* was unchanged (Fig 5F).

Next, we investigated whether the higher plasma magnesium level could explain by itself the decreased serum calcification propensity of *Memo1* cKO mice. We pooled serum samples of 24 individual mice per genotype into 6 independent biological samples and measured the mean difference of magnesium concentration between the two genotypes, which amounted to 0.2 mmol/L. We then spiked serum samples of *Memo1* cKO mice and controls with the same

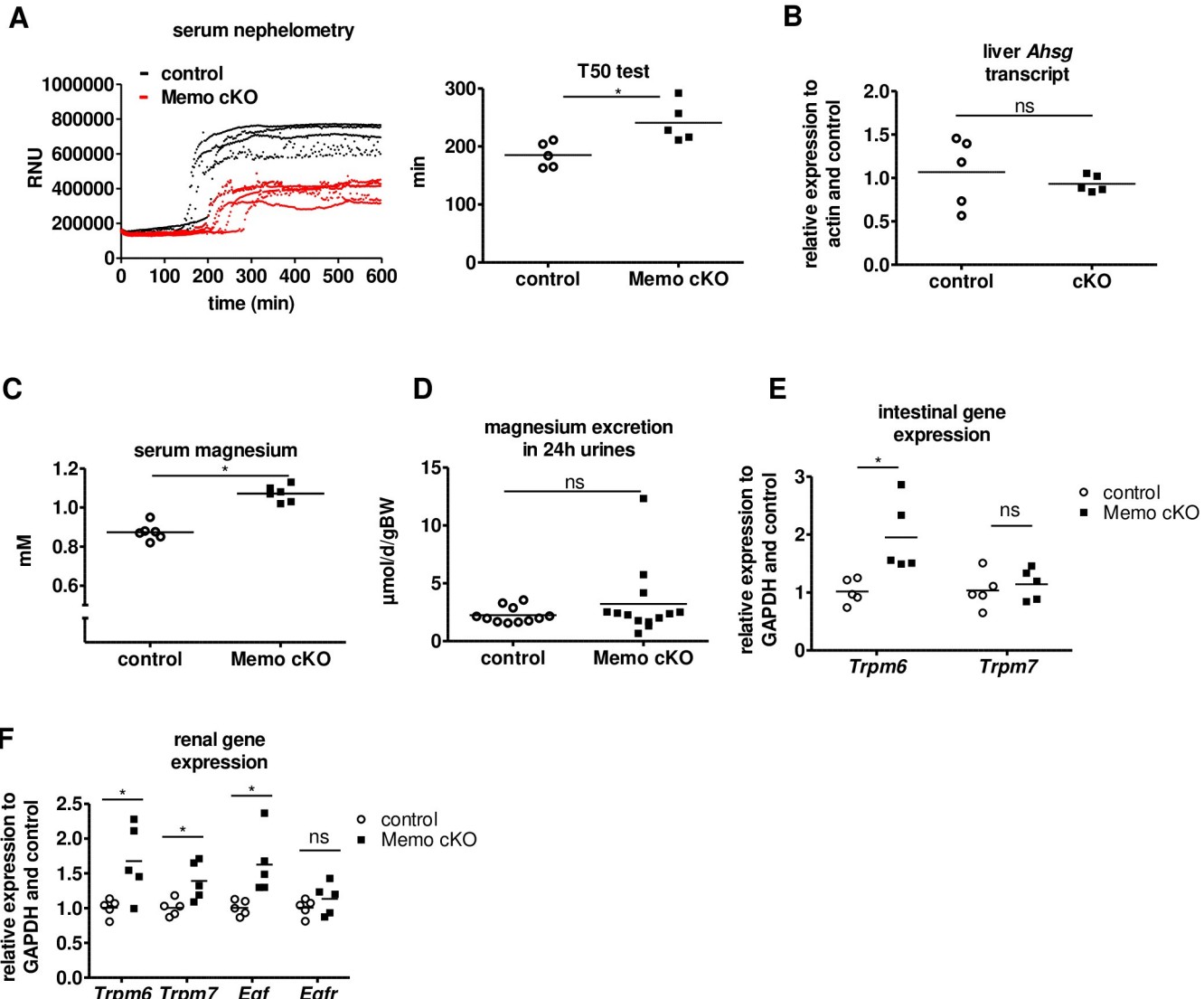

**Fig 5. Altered magnesium homeostasis protects whole-body *Memo1* cKO mice from ectopic mineralization. A**. T50 test. Nephelometric light scattering of spontaneously forming primary and secondary calciprotein particles was measured (left panel). Sera from *Memo1* cKO displayed lower relative nephelometry units (RNU) upon formation of secondary calciprotein particles and longer conversion from primary to secondary particles was increased (right panel), indicating lower propensity to calcify. **B**. Hepatic expression of gene *Ahsg* encoding fetuin-A was comparable between genotypes. **C**. *Memo1* cKO showed higher serum magnesium concentrations than control animals. **D**. Urinary magnesium excretion per 24h was comparable in both genotypes. **E**. In the ileum, *Memo1* cKO mice revealed higher transcription of *Trpm6* than controls, whereas *Trpm7* was unchanged. **F**. In kidney qPCR both *Trpm6* and *Trpm7*, and transcripts of *Egf* were increased in *Memo1* cKO compared to controls. * $p < 0.05$ (t-test); ns, not significant. n = 5 per genotype (A-B, E-F); sera of 24 male and female mice aged 12 weeks pooled to 6 independent samples per genotype (C); 11 controls and 13 *Memo1* cKO (D).

amount of magnesium and measured calcification propensity. This intervention shifted the T50 calcification propensity of spiked control mouse serum to a level comparable to native unspiked serum of *Memo1* cKO (Fig 6). In an experimental control of independently pooled serum from 6 wildtype mice, spiking with 0.2 mmol/L magnesium chloride increased the T50 test from 206 min to 231 min. Taken together, these data indicate that mice are protected from ectopic calcifications by higher serum magnesium resulting from increased expression of magnesium transporting molecules in the kidneys and the gut.

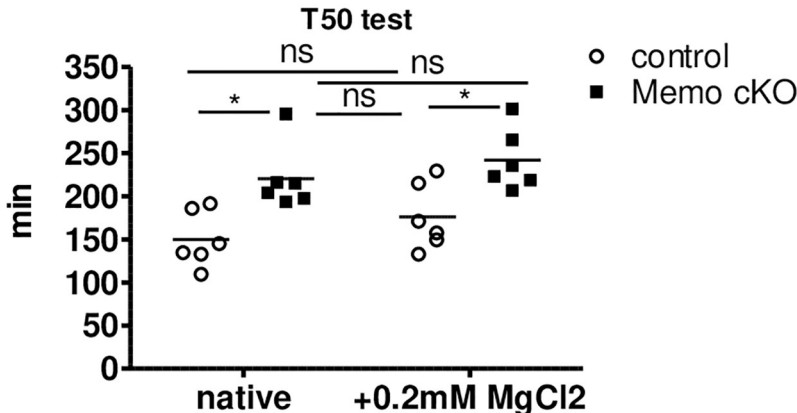

**Fig 6. Difference in serum magnesium levels accounts for altered serum calcification propensity in *Memo1* cKO mice.** Serum samples were analyzed either untreated or after spiking with 0.2mmol/L magnesium chloride, the difference observed in serum magnesium levels between the two genotypes. The resulting T50 test of calciprotein conversion approximated a similar range in spiked sera of control mice when comparing with unspiked *Memo1* cKO. *, p < 0.05 (t-test); ns, not significant. Sera of 24 mice pooled to 6 independent samples per genotype.

## Loss of *Memo1* in the kidney elevates magnesemia but is partially compensated by decreased intestinal magnesium channel expression

Finally, given the essential role of the kidney in magnesium homeostasis, we hypothesized that loss of *Memo1* in the kidney alone could increase renal magnesium reabsorption leading to higher serum magnesium. We have previously established and characterized a model of renal tubule-specific inducible *Memo1* exon 2 deletion using the doxycycline-inducible Pax8-rtTA and LC1 Cre transgenes, as reported elsewhere [25]. Mice were treated with low-dose doxycycline and *Memo1* kKO displayed kidney-specific loss of MEMO1 protein (Fig 7A). Measurement of serum magnesium concentrations in this model, revealed a significantly increased mean value in the *Memo1* kKO mice (1.03 ± 0.04 mmol/L vs 0.97 ± 0.02 mmol/L, p = 0.049) (Fig 7B), but of lower magnitude compared to *Memo1* cKO mice. Fractional excretion of magnesium in the urine was comparable between the two genotypes (Fig 7C).

Probably owing to the small difference in serum magnesium, the *Memo1* kKO mice showed a serum calcification propensity that was comparable to that of control mice (Fig 7D). In this kidney-specific knockout model *Memo1* kKO, we found that ileal expression of *Trpm7* was decreased and that *Trpm6* showed a trend for a decrease in *Memo1* kKO, suggesting a lower magnesium absorption from the intestine (Fig 7E). However, the renal tissue directly affected by the Cre/lox-mediated *Memo1* exon2 excision showed increased *Trpm7* and *Trpm6* expression (Fig 7F). Both renal *Egf* and *Egfr* gene expression were unaffected in this model (Fig 7F). We spiked the small but significant difference in magnesemia between *Memo1* kKO and controls, amounting 0.06mmol/L, (Fig 7D) to serum samples, and we measured the T50 test of serum calcification propensity. No difference was noted between the spiked and unspiked groups (Fig 7D). However, in an experimental control of pooled serum from 6 wildtype mice, adding 0.06 mmol/L magnesium chloride slightly increased the T50 test from 206 min to 214 min.

## No evidence for a direct magnesium-dependent oxidative function of recombinant MEMO1 protein

Since MEMO1 display copper-reducing activity without any known physiological substrate [1], we tested whether MEMO1's redox function might be magnesium-dependent. We

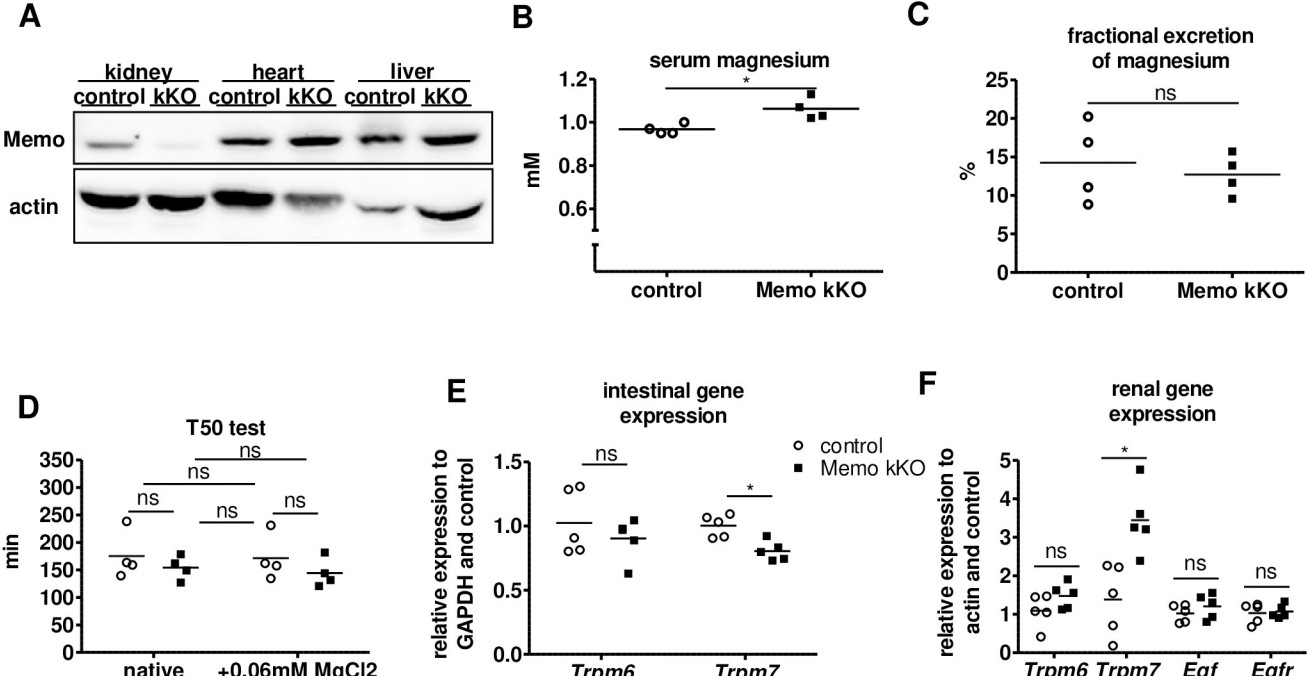

**Fig 7. Kidney-specific *Memo1* kKO mice displayed elevated magnesemia and renal magnesium channel transcription. A**. Renal-specific loss of MEMO1 in *Memo1* kKO was verified using Western blots of whole organ lysates. **B**. Magnesium concentration of sera pooled from 4 independent animals per data point (n = 4x4) was elevated in *Memo1* kKO compared to controls. **C**. Fractional excretion of magnesium was comparable between pools of 4x4 independent murine spot urines per genotype. **D**. In serum pools of (4x4) *Memo1* kKO and controls, T50 test of calciprotein conversion was comparable, and spiking with 0.06 mmol/L magnesium chloride caused no significant change in T50 test of serum calcification propensity. **E**. In ileum, gene expression of *Trpm6* tended to decrease and *Trpm7* significantly decreased in *Memo1* kKO compared to controls. **F**. In the kidney *Trpm6* tended to increase and *Trpm7* significantly increased in *Memo1* kKO, whereas *Egf* and *Egfr* transcription were unchanged. p < 0.05 (t-test); ns, not significant. n = 1 per genotype (A) but representative of several experiments; sera or spot urines of 16 male mice age 8 weeks pooled to 4 independent samples per genotype (B-D); 5 per genotype (E-F).

measured copper-reducing activity of different magnesium chloride concentrations either alone or in presence of 2μg of recombinant MEMO1. While magnesium chloride alone had a slight dose-dependent copper-reducing effect up to 10mM, it did not influence the copper-reducing activity in presence of MEMO1 (Fig 8A). By contrast, magnesium chloride increased the copper-reducing activity of the known antioxidant ascorbic acid that we used as a positive control (Fig 8B).

## Discussion

*Memo1* cKO mice share some characteristics with *Klotho* and *Fgf23*-deficient strains including higher levels of plasma calcium, phosphate and vitamin D which are thought to be causes of increased tissue calcification and premature aging. We assessed whether a decrease in systemic mineral load could prevent premature aging and death in *Memo1* cKO mice. However, neither a low phosphate diet nor a vitamin D deficient diet rescued the survival of *Memo1* cKO mice, which is in contrast to what has been observed in *Klotho* and *Fgf23*-deficient mice.

This illustrates several key differences between the phenotype of *Memo1*- and *Fgf23*- or *klotho*-deficient mice. *Memo1* cKO mice display (i) a distinct bone phenotype due to alkaline phosphatase dysfunction, (ii) absence of soft tissue calcifications and (iii) no rescue by low phosphate or vitamin-D deficient diets. This suggests that *Memo1* cKO mice are resistant to calcification and thus a different mechanism causes their premature aging compared to *Fgf23*-

**A**

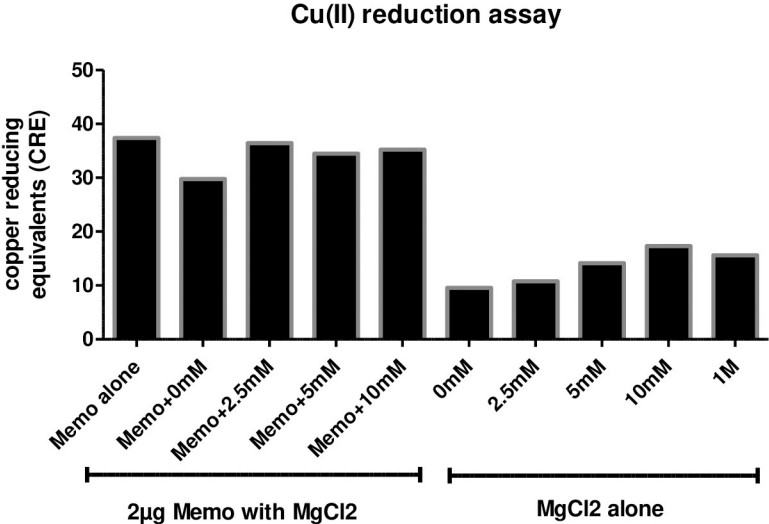

**B**

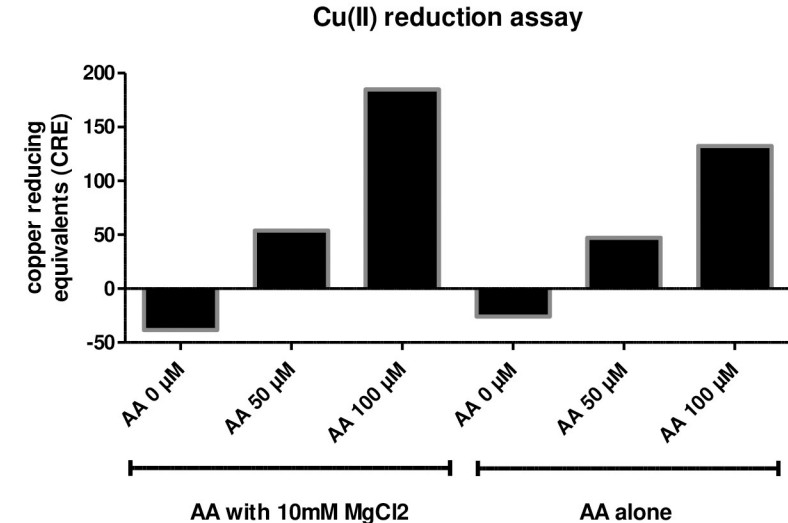

**Fig 8. No evidence for a direct magnesium-dependent oxidative function of recombinant MEMO1 protein. A**.
Addition of indicated concentrations of magnesium chloride did not alter copper-reducing function of recombinant
MEMO1 protein, but magnesium chloride alone showed a dose-dependent copper-reducing effect up to 10mM. **B**.
Magnesium chloride increased the copper-reducing activity of ascorbic acid. All samples were analyzed in duplicates.

or *klotho*-deficient mice. We indeed found that *Memo1* cKO mice are less prone to overall cal-
cification (tissues and serum calcification propensity). We identified higher serum magnesium
concentration in *Memo1* cKO mice as one of the main causes, in the absence of any change in
serum albumin concentration or in liver Fetuin A-coding gene expression (*Ahsg)*, two main
protectors against calcification in the serum.

These results led us to investigate magnesium homeostasis in the *Memo1* cKO and the kidney-specific *Memo1* kKO mouse models. While mice of both models showed higher serum magnesium concentrations, we observed model-specific differences in the tissue-specific regulation of the magnesium transport proteins. We found an increase in the expression of the magnesium channels *Trpm6* and *Trpm7* in the ileum and the kidneys of the whole body MEMO1-deficient mice *Memo1* cKO. Interestingly, kidney-specific MEMO1-deficient *Memo1* kKO mice showed increased *Trpm7* transcripts only in the kidney and a decreased expression in the intestine, suggestive of an intestinal compensation mechanism that keeps plasma magnesium concentration close to the normal range. This may explain the only small elevation of magnesium levels and the similar calcification propensity observed in sera of *Memo1* kKO animals. Overall, we propose that MEMO1 has a role in the regulation of *Trpm6* and *Trpm7* gene expression, as has been previously described for other transport proteins involved in phosphate and calcium homeostasis [2, 25]. Some data are suggestive that phospholipase C has a role in this regulation [26–28], but the precise molecular mechanisms underlying MEMO1regulation of *Trpm7* expression remain unknown.

As MEMO1 participates in multiple cellular functions, some of them only partially understood, the current data reveal a new physiological axis for MEMO1. Although we can only speculate why MEMO1-deficient tissues increase magnesium channels, we propose a combination of different mechanisms that might contribute to explaining our results:

1. MEMO1 might crosslink magnesium and phosphate homeostasis due to its function downstream of both EGFR and FGFR signaling pathways. We have previously shown that MEMO1-deficient cells are resistant to treatment with EGF [4] and FGF23 [2]. In the same line, antibody-mediated EGFR inhibition was recently shown to reduce renal phosphate excretion and concomitantly to decrease magnesemia in mice [29]. MEMO1 may thus be involved in the crosstalk between the two pathways.

2. Regarding phosphate regulation, we have previously shown that *Memo1* cKO mice have decreased renal phosphate transporter expression [2], but preserved phosphatemia. Normal serum phosphate concentration might be explained by the opposite role of MEMO1 on EGFR and FGFR signaling. More specifically, loss of MEMO1 impairs cellular response to FGF23 and decreased NaPi2a and NaPi2c expression [2], thus decreasing renal phosphate excretion, while higher renal *Egf* expression favors phosphate excretion [30, 31]. A decreased renal FGF23 response but increased paracrine EGF activity in the kidney may result, which could explain the normal serum phosphate concentrations in *Memo1* cKO [2, 4].

3. Intriguingly, *Trpm7* which is transcriptionally increased in the two *Memo1* KO mouse models presented here, is needed for EGF-induced cancer cell migration [32], a function apparently opposite to *Memo1*'s role in cancer cell motility, where MEMO1 expression is upregulated [27]. The increased expression of *Trpm7* induced by EGF or by the absence of *Memo1* with opposite role on cancer cell migration will require more in-depth studies.

4. Oxidative dysregulation found in *Memo1*-deficient cells [1] may directly affect the TRPM channel-kinase activity: indeed TRPM6 is suppressed by hydrogen peroxide, and TRPM7 acts as a sensor for mitochondrial oxidative stress [33–36]. In addition, vitamin C synthesis [37] and its transport into tissues via sodium-dependent vitamin C transporter 2 [38] is facilitated by increased magnesium concentrations. This would likely be beneficial for an organism under oxidative stress [39]. Evidence for such physiological compensation mechanisms is however lacking and should be directly assessed in more mechanistic studies.

5. Loss of *Memo1* affects alkaline phosphatase activity in bone and serum [4] and alkaline phosphatase activity depends on magnesium. Thus, an increased magnesium concentration would help rescue alkaline phosphatase activity [40] and overall benefit bones in *Memo1* cKO mice.

It is also important to discuss the limitations of this study. First, the effect of magnesium on serum T50 values might be seen as small compared to what has been previously described [20]. However, the current data stem from C57BL/6 mouse sera, in contrast to previous studies [20] that used mice maintained on the DBA genetic background. Mice with DBA background show lower magnesium concentrations and are more prone to soft-tissue calcifications [41]. Second, *Memo1* cKO mice showed an increase in renal *Egf* mRNA expression, but *Memo1* kKO did not. The comparability of the two models is however limited by the chronic kidney disease (CKD) that develops in *Memo1* cKO [4] but not in *Memo1* kKO mice which have normal serum creatinine levels [2]. Indeed, CKD may itself increase EGF levels [42].

In conclusion, loss of the redox protein MEMO1 alters magnesium homeostasis and, in the case of whole-body MEMO1 depletion, decreases serum calcification propensity. Magnesium protects from vascular calcification in rodents [43] and from progression to end-stage renal disease in hyperphosphatemic patients [44, 45]. In addition, targeting oxidative stress is considered a promising strategy for both acute kidney injury [46–48] and chronic kidney disease [49]. The current data highlight another potential relevance of such strategies in reducing soft tissue calcification, a critical contributor to mortality in kidney disease.

## Supporting information

**S1 File.**
(PDF)

## Acknowledgments

The authors are thankful to Daniel Bardy and his team, Laboratoire Central de Chimie, Lausanne University Hospital for the analysis of mouse serum and urine samples, and to Carsten Wagner and Jürg Biber, University of Zürich, for providing the anti-NaPi2a antibody.

## Author Contributions

**Conceptualization:** Matthias B. Moor, Suresh K. Ramakrishnan, Nancy E. Hynes, Olivier Bonny.

**Data curation:** Matthias B. Moor, Suresh K. Ramakrishnan, Andreas Pasch, Olivier Bonny.

**Formal analysis:** Matthias B. Moor, Suresh K. Ramakrishnan, Olivier Bonny.

**Funding acquisition:** Olivier Bonny.

**Investigation:** Matthias B. Moor, Suresh K. Ramakrishnan, Finola Legrand, Matthias Bachtler.

**Methodology:** Matthias B. Moor, Nancy E. Hynes, Andreas Pasch, Olivier Bonny.

**Project administration:** Olivier Bonny.

**Resources:** Robert Koesters, Olivier Bonny.

**Supervision:** Olivier Bonny.

**Validation:** Matthias B. Moor, Andreas Pasch, Olivier Bonny.

**Writing – original draft:** Matthias B. Moor, Nancy E. Hynes, Olivier Bonny.

**Writing – review & editing:** Matthias B. Moor, Suresh K. Ramakrishnan, Finola Legrand, Matthias Bachtler, Robert Koesters, Nancy E. Hynes, Andreas Pasch, Olivier Bonny.

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
