## [Decision Letter · Decision Letter 0]

6 Apr 2020

PONE-D-20-06005

Elevated serum magnesium lowers calcification propensity in Memo1-deficient mice

PLOS ONE

Dear Dr. Bonny,

I hope this note finds you well. We’re all in a bit of a fog coping with the c19 pandemic. At some level, it seems odd to be concerned with esoteric scientific problems with so much more at stake. That said, I now received reviews on your submitted work from two internationally respected experts. Both found your work of substantial potential interest but raised a number of salient questions and concerns that will have to be favorably addressed before a final decision can be reached.

Reviewer #1 raises the obvious and sensible question regarding the paradoxical increase of Npt2a with elevated Fgf23 levels, when reduced transporter expression would be expected. Moreover, if Npt2a is elevated, what counterregulatory mechanism keeps serum Pi normal?

Reviewer #2 expressed greater reservations and questions whether MEMO1 deficiency causes such profound pathology that the phenotype arises secondarily. The reviewer also notes the apparent contradictory findings with Fgf23 and Npt2a, along with other significant concerns. Some suggestions for addressing these perceived deficiencies are included by both reviewers.

The journal looks forward to your response and an amended manuscript.

We would appreciate receiving your revised manuscript by May 21 2020 11:59PM. To enhance the reproducibility of your results, we recommend that if applicable you deposit your laboratory protocols in protocols.io, where a protocol can be assigned its own identifier (DOI) such that it can be cited independently in the future. For instructions see: http://journals.plos.org/plosone/s/submission-guidelines#loc-laboratory-protocols

We look forward to receiving your revised manuscript.

With kind regards,

Peter A. Friedman, Ph.D. (a former Institut fellow)

Academic Editor

PLOS ONE

Journal Requirements:

"is an inventor of the T50-Test and stock-holder and employee of Calciscon AG, which commercialized the T50-Test (calcification propensity test). MB is an employee of Calciscon AG. The other authors declare that no conflict of interest exists."

5. Your ethics statement must appear in the Methods section of your manuscript. If your ethics statement is written in any section besides the Methods, please move it to the Methods section and delete it from any other section. Please also ensure that your ethics statement is included in your manuscript, as the ethics section of your online submission will not be published alongside your manuscript.

6. Thank you for stating the following in the Financial Disclosure section:

"The study was sponsored by the Swiss National Science Foundation through the program NCCR-Kidney.CH and by an unrestricted grant from the patient association “Association pour l’Information et la Recherche sur les maladies rénales Génétiques (AIRG)-Suisse”."

We note that one or more of the authors are employed by a commercial company: Calciscon AG.

Reviewers' comments:

Reviewer's Responses to Questions

**Comments to the Author**

1. Is the manuscript technically sound, and do the data support the conclusions?

Reviewer #1: Yes

Reviewer #2: Partly

2. Has the statistical analysis been performed appropriately and rigorously? 

Reviewer #1: Yes

Reviewer #2: Yes

3. Have the authors made all data underlying the findings in their manuscript fully available?

Reviewer #1: Yes

Reviewer #2: Yes

4. Is the manuscript presented in an intelligible fashion and written in standard English?

Reviewer #1: Yes

Reviewer #2: Yes

5. Review Comments to the Author

Reviewer #1: The manuscript by Moor et al describes a potential mechanism for the reduction of calcifications in Memo-1 deficient mice. The authors’ work suggests this could be due to a role of Memo-1 in magnesium handling. Overall, the work adequately supports the hypothesis, however several clarifications should be made:

1) It is not clear why in the cKO mouse that if FGF23 is elevated, why is NPT2a increased, when a principle action of FGF23 is to down-regulate this protein? Further, why are there conflicting data with the 24-OHase mRNA levels? It appears as if this gene is properly regulated by the elevated FGF23, as it shows increased expression.

2) Would suggest that the mice be re-assessed with the Quidel Rodent specific FGF23 intact ELISA, not the Kainos human FGF23 ELISA that was used previously.

3) If NPT2a is elevated, what is the counter regulatory mechanism to keep the serum Pi normal in the cKO mice since 1,25D and PTH are normal?

4) It is clear that FGF23 and KL are genes involved in phosphate handling, and are not true aging-related genes as in progeria. References to aging-like phenotypes in the cKO mouse model should be minimized in the Introduction and a sentence should be added that these phenotypes can be explained by altered phosphate and vitamin D handling.

5) If possible, in vitro experiments would strengthen the idea that Memo-1 directly controls Trpm6-7 expression.

Reviewer #2: Mediator of cell ErbB2-driven MOtility1 (MEMO1) is a ubiquitously expressed protein with vital functions. Structurally, MEMO1 is related to iron- and zinc-binding enzymes, and exerting a Cu-II dependent oxidase activity affecting the cellular redox status and controlling the activity of a wide range of redox-regulated cellular proteins. MEMO1 affects cellular motility presumably by direct interaction with cellular motility proteins, and in addition through scaffolding of signalosomes. Thus MEMO1 exhibits profound protein moonlighting, which renders in vivo studies near impossible because or complex interaction and interdependencies of the claimed functions. Whole body MEMO1 knockout is embryo lethal, while conditional knockout cKO produces a complex phenotype of growth retardation and drastically reduced life span (mean survival about 2 months). These findings suggest that MEMO1 deficiency renders the animals profoundly sick and that it will be very difficult to decide if aspects of the phenotype are the cause or a consequence of the progressively poorer health status of the animals, again begging the question, which of the many claimed functions of MEMO1 is chief in this scenario. Studies suggested that a major function of MEMO1 is enhancing FGF signaling by stabilizing the FGFR signalosome. This broad function downstream of receptor tyrosine kinases was narrowed down when it was realized that the MEMO1 cKO phenotype was similar to the phenotype of FGF23 and klotho deficient mice. This finding put MEMO1 into FGF23-klotho phosphaturic signaling, however with one major shortcoming, namely that the overt hyperphosphatemia of FGF23 and klotho knockout mice that is meant to be instrumental for a large part of the detrimental knockout phenotypes is completely absent in MEMO1 cKO. One driving feature of FGF23 and klotho knockout morbidity is pathological calcification, which can be (partially) rescued by low phosphate diet or by reducing Vit D activity in the mice. Against this background the authors study calcification in MEMO1 cKO mice and unlike in FGF23 mice, find none. Dietary phosphate or vitamin-D restriction do not change the survival of the MEMO1 cKO mice. The authors report however, slight differences in serum calcification propensity, that they attribute to serum Mg.

The amount and quality of work presented in the manuscript make it well worth reporting even if it were purely descriptive. In the current mechanistic presentation I have great difficulties following the authors’ reasoning, especially that a slight change in serum magnesium will affect the health of the animals so profoundly.

Major points

1) It would help a lot if the authors would lay out their working hypothesis more clearly given the many purported functions of MEMO1, especially since some of the crucial disturbances in mineral handling observed in FGF23 and klotho mice are completely absent in MEMO1 cKO mice, or minor at best.

Minor points

1) Hyperphosphatemia and calcification are absent in the mice, why go on and study calcification propensity? Please explain.

2) Experimental diets; 0.65% phosphorous in the control diet is still pretty high, 0.8% definitely is. For comparison see a recent study by Babler et al. 2020 PLoS ONE (https://doi.org/10.1371/journal.pone.0228938)

3) The differences in control serum magnesium in Fig 5C and Fig 7B are almost as large like the differences between control and kKO. Please discuss.

4) Fig 4F x-rays are pretty bad quality. Since the authors used a Skyscan micro-CT, can they provide higher resolution CT scans? Perhaps these would show microcalcifications.

5) Fig 7 and the function of Mg transporters in various tissues, please discuss against a recent publication on this topic (Chubanov et al. 2016, doi.org/10.7554/eLife.20914)

6) Fig 9 does not add any information to the text, delete.

6. PLOS authors have the option to publish the peer review history of their article (what does this mean?). If published, this will include your full peer review and any attached files.

Reviewer #1: No

Reviewer #2: Yes: Willi Jahnen-Dechent

---

## [Author Response · Author response to Decision Letter 0]

23 Jun 2020

REPLY to the EDITOR and the REVIEWERS: available in a separate document in FULL. Here is only a short version of it:

Dear Dr. Bonny,

I hope this note finds you well. We’re all in a bit of a fog coping with the c19 pandemic. At some level, it seems odd to be concerned with esoteric scientific problems with so much more at stake. That said, I now received reviews on your submitted work from two internationally respected experts. Both found your work of substantial potential interest but raised a number of salient questions and concerns that will have to be favorably addressed before a final decision can be reached.

Reviewer #1 raises the obvious and sensible question regarding the paradoxical increase of Npt2a with elevated Fgf23 levels, when reduced transporter expression would be expected. Moreover, if Npt2a is elevated, what counterregulatory mechanism keeps serum Pi normal?

Reviewer #2 expressed greater reservations and questions whether MEMO1 deficiency causes such profound pathology that the phenotype arises secondarily. The reviewer also notes the apparent contradictory findings with Fgf23 and Npt2a, along with other significant concerns. Some suggestions for addressing these perceived deficiencies are included by both reviewers.

Dear Dr Friedman,

Many thanks for your kind words. I also hope that you are doing fine. Apart of treating covid19 patients, we also found the time to address the different points raised by the two reviewers and thank them for their interesting and constructive comments. In accordance with the pointed issues, we changed the text when needed.

We hope that the manuscript will be now acceptable for the readership of PLoSOne.

Best regards

Olivier Bonny, in the name of the co-authors

Response to the reviewers:

Reviewer #1: The manuscript by Moor et al describes a potential mechanism for the reduction of calcifications in Memo-1 deficient mice. The authors’ work suggests this could be due to a role of Memo-1 in magnesium handling. Overall, the work adequately supports the hypothesis, however several clarifications should be made:

1) It is not clear why in the cKO mouse that if FGF23 is elevated, why is NPT2a increased, when a principle action of FGF23 is to down-regulate this protein? Further, why are there conflicting data with the 24-OHase mRNA levels? It appears as if this gene is properly regulated by the elevated FGF23, as it shows increased expression.

We agree and are also puzzled by the differential regulation of NaPi2a and 1,25-D in the Memo1 KO mouse model. We have no definitive mechanistic explanations for this observation based on in vivo data. We can only speculate that FGF23-driven signaling is not regulating NPT2a as expected (as previously shown by Haenzi B et al., FASEB J, 2014), and hence NPT2a mRNA is not repressed in Memo1-cKO mice kidney. One possible explanation suggested by Haenzi et al regards the processing of NPT2a. Haenzi et al have found that the quantity of the clived 40KDa fragment of NPT2a was reduced in Memo1 KO mice renal BBMVs. However, this was confirmed neither in the present work nor in the kidney-specific Memo1-KO mouse model (Moor et al, Front in Physiol). Moreover, the significance of NPT2a cleavage is unknown to the best of our knowledge. 

It should also be pointed out here that NPT2c gene expression, by contrast, is consistently decreased in all Memo1 KO mouse models and thus responds the expected way to FGF23.

The 24-OHase mRNA increase in Memo1 KO mice is associated with increased FGF23 in Memo whole-body KO but was not found in kidney-specific KO animals (Moor et al., Front Physiol 2018), not suggesting a causal relationship of this finding in the present study.

Overall, it seems that Memo1 is regulating transcriptionally and/or post-translationally some specific transporters involved in mineral metabolism, including NPT2a, but without affecting phosphate homeostasis. As a reminder, data on calcium transporters indicate that the absence of Memo1 changes the transcriptional and post-translational expression pattern of NCX1, TRPV5 and calbindinD28k.

Further in vitro studies need to be performed in order to explore the apparent contradiction on the regulation of phosphate co-transporters and vitamin D in these mice but are beyond the scope and message of this paper. A dedicated manuscript addressing FGF23 regulation in Memo1 cKO mice is in preparation.

2) Would suggest that the mice be re-assessed with the Quidel Rodent specific FGF23 intact ELISA, not the Kainos human FGF23 ELISA that was used previously. 

We thank this reviewer for this suggestion and will consider a mouse kit for future studies. This was not available to our knowledge when the current study was conducted. In that period the human Kainos kit was standard in the field, and the kit is validated for use in mouse, rat and monkey according to the manufacturer.

3) If NPT2a is elevated, what is the counter regulatory mechanism to keep the serum Pi normal in the cKO mice since 1,25D and PTH are normal? 

This is an intriguing point that remains unsolved so far. We have not performed detailed phosphate balance and have no data on intestinal absorption and proximal tubule phosphate absorption. Additionally, Memo1-KO mice have impaired bone mineralization that may contribute to keep phosphatemia in the normal range (Moor et al, JBMR Plus 2018). As discussed previously, the discrepancy in the expression of NaPi2a (increased in BBMV) and NaPi2c (mRNA decreased) may also partially compensate, even if both phosphate transporters have distinct transport characteristics and that NPT2C KO mice have no obvious phosphate phenotype.

4) It is clear that FGF23 and KL are genes involved in phosphate handling, and are not true aging-related genes as in progeria. References to aging-like phenotypes in the cKO mouse model should be minimized in the Introduction and a sentence should be added that these phenotypes can be explained by altered phosphate and vitamin D handling. 

As developed in the paper by Haenzi et al (FASEB J, 2014), Memo1 cKO mice do display several traits evoking premature aging not directly related to possible alteration of the mineral metabolism: alopecia, grey hairs, subcutaneous fat atrophy, gonad atrophy, renal insufficiency and insulin resistance. 

Moreover, Kl KO mice have a more complex phenotype than just alteration of the mineral metabolism that includes also anti-oxidant properties (Brobey RK et al, PLoS One. 2015).

In order to follow on the comment of the reviewer, we modified the introduction and abstract as requested by adding a sentence stating that Kl and Fgf23 KO mice have disturbances of the mineral metabolism as major trigger for the observed phenotype.

5) If possible, in vitro experiments would strengthen the idea that Memo-1 directly controls Trpm6-7 expression. 

Memo1 overexpression was lethal in the laboratory of co-author Nancy Hynes, hence we decided not to pursue co-expression studies that would address this point. In addition, there is no association between MEMO1 and TRPM6 expression in RNAseq of 1457 cell lines in Cancer Cell Line Encyclopedia, see illustration below.

Reviewer #2: Mediator of cell ErbB2-driven MOtility1 (MEMO1) is a ubiquitously expressed protein with vital functions. Structurally, MEMO1 is related to iron- and zinc-binding enzymes, and exerting a Cu-II dependent oxidase activity affecting the cellular redox status and controlling the activity of a wide range of redox-regulated cellular proteins. MEMO1 affects cellular motility presumably by direct interaction with cellular motility proteins, and in addition through scaffolding of signalosomes. Thus MEMO1 exhibits profound protein moonlighting, which renders in vivo studies near impossible because of complex interaction and interdependencies of the claimed functions. Whole body MEMO1 knockout is embryo lethal, while conditional knockout cKO produces a complex phenotype of growth retardation and drastically reduced life span (mean survival about 2 months). These findings suggest that MEMO1 deficiency renders the animals profoundly sick and that it will be very difficult to decide if aspects of the phenotype are the cause or a consequence of the progressively poorer health status of the animals, again begging the question, which of the many claimed functions of MEMO1 is chief in this scenario. Studies suggested that a major function of MEMO1 is enhancing FGF signaling by stabilizing the FGFR signalosome. This broad function downstream of receptor tyrosine kinases was narrowed down when it was realized that the MEMO1 cKO phenotype was similar to the phenotype of FGF23 and klotho deficient mice. This finding put MEMO1 into FGF23-klotho phosphaturic signaling, however with one major shortcoming, namely that the overt hyperphosphatemia of FGF23 and klotho knockout mice that is meant to be instrumental for a large part of the detrimental knockout phenotypes is completely absent in MEMO1 cKO. One driving feature of FGF23 and klotho knockout morbidity is pathological calcification, which can be (partially) rescued by low phosphate diet or by reducing Vit D activity in the mice. Against this background the authors study calcification in MEMO1 cKO mice and unlike in FGF23 mice, find none. Dietary phosphate or vitamin-D restriction do not change the survival of the MEMO1 cKO mice. The authors report however, slight differences in serum calcification propensity, that they attribute to serum Mg.

The amount and quality of work presented in the manuscript make it well worth reporting even if it were purely descriptive. In the current mechanistic presentation I have great difficulties following the authors’ reasoning, especially that a slight change in serum magnesium will affect the health of the animals so profoundly. 

We thanks this reviewer for his perfect summary of the knowledge about Memo1’s role in physiology. We agree that the phenotype of cKO mice is complex and might be influenced by the rapid decline of mice health and many confounding factors. This is precisely the reason why we performed the experiments early after KO induction in order to avoid most of the confounding factors.

We must state here as well that when we started this project, we had data from Haenzi et al, FASEB J showing that MEMO1 cKO had high 1,25-D and hypercalcemia. At that time, ectopic tissue calcifications were not looked at. Even if phosphatemia was normal, higher plasma calcium and calcitriol level were highly suggestive for possible ectopic calcification as being central to premature aging. We thus performed the low phosphate and low D diet experiments. We noticed that Memo1 cKO mice that have been backcrossed on a pure C57BL/6 genetic background displayed normal levels of vitamin D, similar to Klotho mice that were backcrossed to C57BL/6 where variants in Cyp24a1 gene were responsible for this discrepancy between mouse strains (Singh A, Aging Cell 2019). As serum calcium and phosphate homeostasis are tightly regulated, we hypothesized that subtle changes in calcium and phosphate fluxes might still take place and explain the phenotype and look for surrogate markers of calcification propensity. We also noted that all distal calcium transporters are upregulated in Memo1 cKO mice, but the slightly elevated calcemia is not exceeding the normal range, probably due to compensation mechanisms taking place in the bone/gut/other tubular segments.

Regarding the effect of magnesium on calcification propensity, we remind here that Pasch et al have clearly shown that even small addition of magnesium (spike with 0.25mmol) did modify the T50 substantially (Pasch et al, JASN 2012). 

Even if Memo1 cKO mice have no hyperphosphatemia or major alterations of mineral metabolism, we are still impressed by the lower T50 present in these mice and the higher serum magnesium concentration. This is the main finding of this study that started from another perspective.

Major points

1) It would help a lot if the authors would lay out their working hypothesis more clearly given the many purported functions of MEMO1, especially since some of the crucial disturbances in mineral handling observed in FGF23 and klotho mice are completely absent in MEMO1 cKO mice, or minor at best.

We now modified the introduction, and more specifically the paragraph presenting the hypothesis at the end of the introduction.

Minor points

1) Hyperphosphatemia and calcification are absent in the mice, why go on and study calcification propensity? Please explain.

Calcification propensity assess more than hyperphosphatemia and tissue calcification. It reflects the long term risk of calcification. And long term is not appreciable in Memo1 KO mice, as they live only about 2 months at best. This is the reason why we look for a surrogate marker of the long-term calcification risk in these mice and collaborated with Dr A. Pasch to measure calcification propensity.

2) Experimental diets; 0.65% phosphorous in the control diet is still pretty high, 0.8% definitely is. For comparison see a recent study by Babler et al. 2020 PLoS ONE (https://doi.org/10.1371/journal.pone.0228938)

Literature is full of diverse diets and we could find a broad spectrum of so-called normal diets or phosphate-rich /phosphate low diets. In our animal facility, standard diet contains 0.83% of phosphate (KLIBA, Kaiseraugst, Switzerland, diet #3800 for standard mouse breeding) and experimental diets 0.2% (low) and 0.8% (normal) phosphate. Phosphate-enriched diets usually contains 1.2% phosphate or more in C57 genetic background (doi: 10.1371/journal.pone.0177942).

In Babler’s paper, the regular Snniff chow diet (1535) contains 0.7% phosphorus and experimental mice were exposed to 0.2 , 0.4 or 0.8% phosphate. Of special note, those mice had a special genetic background (D2) that predispose them to calcification at lower phosphate diets.

3) The differences in control serum magnesium in Fig 5C and Fig 7B are almost as large like the differences between control and kKO. Please discuss. 

There is a substantial difference, as the reviewer points out. This might be explained by differences in gender, age and genetic background of the mice. Data points in 5C each contain serum from 2 males and 2 females aged around 12 weeks and that underwent treatment with tamoxifen. Data points in 7B are males only aged 8 weeks and underwent treatment with doxycycline. In addition, the mouse strains are separated by at least 5 breeding generations. Hence, several variables limit the comparison of different control animal groups.

The gender and age of the mice have been added to the figure legend.

4) Fig 4F x-rays are pretty bad quality. Since the authors used a Skyscan micro-CT, can they provide higher resolution CT scans? Perhaps these would show microcalcifications.

Whole-mouse CT scan with the setup (1076 Skyscan) is difficult to perform, since it would require around 24 hours of scanning per sample. In addition, we do not currently have access to the facility. When dissecting the mice and by light microscope, we did not identify any area with crystal deposition.

5) Fig 7 and the function of Mg transporters in various tissues, please discuss against a recent publication on this topic (Chubanov et al. 2016, doi.org/10.7554/eLife.20914)

In their paper, Chubanov et al. claim that TRPM6 expression in the kidney does not affect the overall Mg homeostasis. They used TRPM6 floxed mice crossed with Ksp-cre. We know the ksp-cre transgenic mouse model very well in the lab and know that this model leads to patchy and incomplete deletion of the gene of interest especially in the DCT (Lee HW et al, AJP 2009; Grahammer F et al, JCI, 2016; ). We do not believe that our data are contrasting with the data of Chubanov’s publication, but simply that Chubanov et al have only a partial KO of TRPM6 resulting in normal magnesemia. TRPM6 is well established as involved in magnesium homeostasis.

6) Fig 9 does not add any information to the text, delete.

 We deleted Figure 9.

---

## [Decision Letter · Decision Letter 1]

7 Jul 2020

Elevated serum magnesium lowers calcification propensity in Memo1-deficient mice

PONE-D-20-06005R1

Dear Dr. Bonny,

We’re pleased to inform you that your manuscript has been judged scientifically suitable for publication and will be formally accepted for publication once it meets all outstanding technical requirements.

Kind regards,

Peter A. Friedman, Ph.D.

Academic Editor

PLOS ONE

Additional Editor Comments (optional):

Reviewers' comments:

Reviewer's Responses to Questions

**Comments to the Author**

1. If the authors have adequately addressed your comments raised in a previous round of review and you feel that this manuscript is now acceptable for publication, you may indicate that here to bypass the “Comments to the Author” section, enter your conflict of interest statement in the “Confidential to Editor” section, and submit your "Accept" recommendation.

Reviewer #1: All comments have been addressed

Reviewer #2: All comments have been addressed

2. Is the manuscript technically sound, and do the data support the conclusions?

Reviewer #1: (No Response)

Reviewer #2: Yes

3. Has the statistical analysis been performed appropriately and rigorously? 

Reviewer #1: (No Response)

Reviewer #2: N/A

4. Have the authors made all data underlying the findings in their manuscript fully available?

Reviewer #1: (No Response)

Reviewer #2: Yes

5. Is the manuscript presented in an intelligible fashion and written in standard English?

Reviewer #1: (No Response)

Reviewer #2: Yes

6. Review Comments to the Author

Reviewer #1: (No Response)

Reviewer #2: (No Response)

7. PLOS authors have the option to publish the peer review history of their article (what does this mean?). If published, this will include your full peer review and any attached files.

Reviewer #1: No

Reviewer #2: **Yes: **Willi Jahnen-Dechent

---

## [Editor Report · Acceptance letter]

10 Jul 2020

PONE-D-20-06005R1 

Elevated serum magnesium lowers calcification propensity in Memo1-deficient mice 

Dear Dr. Bonny:

I'm pleased to inform you that your manuscript has been deemed suitable for publication in PLOS ONE. Congratulations! Your manuscript is now with our production department. 

Kind regards, 

on behalf of

Dr. Peter A. Friedman 

Academic Editor

PLOS ONE